# Inflammation and Venous Thromboembolism in Hospitalized Patients with COVID-19

**DOI:** 10.3390/diagnostics13223477

**Published:** 2023-11-19

**Authors:** Angelos Liontos, Dimitrios Biros, Rafail Matzaras, Konstantina-Helen Tsarapatsani, Nikolaos-Gavriel Kolios, Athina Zarachi, Konstantinos Tatsis, Christiana Pappa, Maria Nasiou, Eleni Pargana, Ilias Tsiakas, Diamantina Lymperatou, Sempastien Filippas-Ntekouan, Lazaros Athanasiou, Valentini Samanidou, Revekka Konstantopoulou, Ioannis Vagias, Aikaterini Panteli, Haralampos Milionis, Eirini Christaki

**Affiliations:** 11st Division of Internal Medicine & Infectious Diseases Unit, University General Hospital of Ioannina, Faculty of Medicine, University of Ioannina, 45500 Ioannina, Greece; angelosliontos@gmail.com (A.L.); dimitrisbiros@gmail.com (D.B.); rafail.matz@gmail.com (R.M.); il.tsiakas@gmail.com (I.T.); andalimperatou@gmail.com (D.L.); sebastienfilippas@gmail.com (S.F.-N.); lazathanasiou@gmail.com (L.A.); valentinasmnsmn@gmail.com (V.S.); revekkakon@gmail.com (R.K.); doc_gvagias@yahoo.gr (I.V.); katerinapanteli@hotmail.com (A.P.); hmilioni@uoi.gr (H.M.); 2School of Electrical and Computer Engineering, National Technical University of Athens, 15773 Athens, Greece; ktsarapatsani@gmail.com; 3Faculty of Medicine, University of Ioannina, 45110 Ioannina, Greece; ngkolios99@gmail.com (N.-G.K.); christianapappa4@gmail.com (C.P.); md06549@uoi.gr (M.N.); md06655@uoi.gr (E.P.); 4Department of Otorhinolaryngology, Head and Neck Surgery, University General Hospital of Ioannina, Faculty of Medicine, University of Ioannina, 451100 Ioannina, Greece; athinazarachi@gmail.com; 5Department of Respiratory Medicine, University General Hospital of Ioannina, Faculty of Medicine, University of Ioannina, 451100 Ioannina, Greece; konstantatsis@gmail.com

**Keywords:** COVID-19, inflammatory indices, venous thromboembolism, interleukin-6, lymphocytopenia, C-reactive protein, neutrophil-to-lymphocyte ratio, triglyceride–glucose index

## Abstract

Background: A link between inflammation and venous thromboembolism (VTE) in COVID-19 disease has been suggested pathophysiologically and clinically. The aim of this study was to investigate the association between inflammation and disease outcomes in adult hospitalized COVID-19 patients with VTE. Methods: This was a retrospective observational study, including quantitative and qualitative data collected from COVID-19 patients hospitalized at the Infectious Diseases Unit (IDU) of the University Hospital of Ioannina, from 1 March 2020 to 31 May 2022. Venous thromboembolism was defined as a diagnosis of pulmonary embolism (PE) and/or vascular tree-in-bud in the lungs. The burden of disease, assessed by computed tomography of the lungs (CTBoD), was quantified as the percentage (%) of the affected lung parenchyma. The study outcomes were defined as death, intubation, and length of hospital stay (LoS). A chi-squared test and univariate logistic regression analyses were performed in IBM SPSS 28.0. Results: After propensity score matching, the final study cohort included 532 patients. VTE was found in 11.2% of the total population. In patients with VTE, we found that lymphocytopenia and a high neutrophil/lymphocyte ratio were associated with an increased risk of intubation and death, respectively. Similarly, CTBoD > 50% was associated with a higher risk of intubation and death in this group of patients. The triglyceride–glucose (TyG) index was also linked to worse outcomes. Conclusions: Inflammatory indices were associated with VTE. Lymphocytopenia and an increased neutrophil-to-lymphocyte ratio negatively impacted the disease’s prognosis and outcomes. Whether these indices unfavorably affect outcomes in COVID-19-associated VTE must be further evaluated.

## 1. Introduction

Coronavirus disease 2019 (COVID-19) is caused by severe acute respiratory syndrome coronavirus-2 (SARS-CoV-2) [1]. From the beginning of the pandemic to 21 July 2023, there were 768,237,788 confirmed COVID-19 cases, including 6,951,677 deaths worldwide [2]. However, it has been estimated that excess mortality attributed to COVID-19 may be much higher [2].

Clinically, patients with COVID-19 exert manifestations of the disease ranging from asymptomatic illness to critical forms [3,4]. SARS-CoV-2 primarily targets the respiratory tract [5]. Viral entry into cells is mediated by the angiotensin-converting enzyme-2 (ACE-2) receptors of the nasal and alveolar endothelium [5,6]. Histopathological findings in COVID-19 patients showed diffuse alveolar damage and inflammatory infiltrates of the lung parenchyma [7,8,9,10]. Similarly, such infiltrates were found in extra-pulmonary sites, such as the gastrointestinal tract and cardiac muscle [7,8,9,10]. During the initial phase of the pandemic, it was estimated that 3–10% of infected patients would require hospitalization owing to progressive pneumonia; of those, 20% would develop a more severe or critical form of the disease, namely acute respiratory distress syndrome (ARDS), although the clinical course may be dependent on the SARS-CoV-2 variant [11,12].

Severe COVID-19 disease is associated with an exaggerated host immune response mediated by chemotaxis [13]. This overt inflammatory state and immune over-activation, in some individuals, can induce cytokine release syndrome [14]. In this syndrome, the levels of circulating inflammatory mediators are found to be elevated (i.e., interleukin-6 (IL-6), interleukin- 1β (IL-1β), interleukin-2 (IL-2), interleukin-6 (IL-6), interleukin-7 (IL-7), interleukin-10 (IL-10), interleukin-18 (IL-18), interferon gamma-induced protein 10 (IP-10), monocyte chemoattractant protein-1 (MCP-1), tumor necrosis factor-alpha (TNF-α), macrophage inflammatory protein 1 alpha (MIP-1α), granulocyte colony-stimulating factor (G-CSF)) while lymphocyte levels are decreased [13,15,16,17,18]. Various markers have been used as indices of the inflammatory process and severity of the disease’s course [19]. Among these, low absolute counts of lymphocytes and platelets, as well as elevated levels of cardiac troponin, ferritin, IL-6, and the neutrophil-to-lymphocyte ratio, have been associated with worsening disease and unfavorable outcomes [13,15,17,18,19,20,21,22,23].

The host immune response after infection with a variety of pathogens can lead to thrombi formation, particularly in microvessels [24,25]. Similarly, COVID-19-associated coagulopathy and hypercoagulation were identified early in the pandemic as emerging complications of the infection resulting in an increased risk for arterial and venous thromboembolism (VTE) [26,27,28]. Patients with severer forms of the disease exert a viral-induced prothrombotic state manifested by the formation of microthrombi or larger vessel thrombosis owing to uncontrolled immunothrombosis [29]. SARS-CoV-2 directly damages the endothelium and activates the coagulation pathway [30,31,32,33]. In addition, the virus, through numerous other pathways, ignites the immunothrombosis cascade, resulting in a thrombus formation in various vascular beds [29,34].

Venous thromboembolic events in the acute phase of COVID-19, including deep vein thrombosis (DVT), primarily of the lower extremities, and pulmonary embolism (PE) are considered common complications [35,36,37,38]. Another entity, namely vascular tree-in-bud (VTIB), was primarily described as thrombotic microangiopathy in pulmonary tumors [39]. Similarly, this form of pulmonary microthrombosis has been identified in COVID-19 patients with the use of computed tomography (CT) of the lungs [40].

In the field of healthcare, algorithms based on machine learning (ML) have been shown to be useful tools with great promise, especially when it comes to predicting clinical events and results, as demonstrated during the COVID-19 pandemic. ML is a branch of artificial intelligence that creates computer programs that can carry out activities that typically require human intelligence [41]. The goal of the popular field of machine learning technology is to create a computer system that can mimic human intelligence. Researchers and medical doctors may extract insightful information and make well-informed decisions by using these models, which have an important role in leveraging the large and intricate datasets produced by medical centers, even in databases with a small sample size [42,43,44]. Despite the risk of bias with smaller datasets, ML models have assisted in the prediction of serious outcomes in COVID-19, such as the progress of disease, deaths, and hospitalization [45,46,47].

Data regarding the association between markers of inflammation and VTE occurrence in COVID-19 are limited. In addition, only a few clinical studies have assessed the role of these markers in COVID-19-related VTE outcomes. The aim of this study was to investigate the association between inflammation and outcomes in hospitalized COVID-19 patients with VTE.

## 2. Materials and Methods

### 2.1. Study Design and Data Extraction

This was a retrospective study. All patients included in this study were admitted to the Infectious Diseases Unit of the University Hospital of Ioannina from 1 March 2020 to 31 May 2022. The patients’ quantitative and qualitative data were obtained from hospital medical records (hard copies and digital records) on epidemiological, clinical, and laboratory parameters. Laboratory data were acquired upon admission. Diagnoses of SARS-CoV-2 infection were confirmed by reverse transcriptase–polymerase chain reaction (RT-PCR) tests on nasopharyngeal swab specimens. We included hospitalized patients aged ≥18 years, with a positive RT-PCR test, independently of COVID-19 disease severity. Patients with missing data on outcomes were excluded. This study is part of a larger hospitalized COVID-19 patient cohort study, which was approved by the Institutional Ethics Committee of the University Hospital of Ioannina (Protocol Number: 5/11-03-2021 (issue: 3)/The University Hospital of Ioannina COVID-19 Registry, NCT05534074).

All data were collected in agreement with the higher standards as set by the respective European Guidelines for Good Clinical and Laboratory Practice in Research Studies/Protocols and in accordance with the Helsinki Declaration. The patients’ medical records were anonymized and imported into a digital database. A unique personal identifier code was used for each patient, as prespecified by the study protocol, and kept anonymous. All data collected and stored from each patient were linked only to this code. Biological samples were not collected. Data regarding patient demographics, anthropometric characteristics, medical history, comorbidities, and concomitant medications were documented upon admission (baseline characteristics). All data were archived in electronic password-encrypted databases. A patient consent form was waived due to the retrospective study design and the anonymization of the database that was used.

A CT pulmonary angiogram or chest CT was used to obtain radiological findings and indices. Radiographic confirmation of PE or VTIB in either exam was considered as evidence of VTE. The CT burden of disease (CTBoD) was quantified as the percentage (%) of the affected lung parenchyma. Lung involvement was assessed using a methodology similar to that in the study by Chung et al. [48]. Vascular ultrasonography was not routinely available in the COVID-19 wards or intensive care units (ICU), and thus, clinically suspected DVT could not be definitively confirmed. In addition, patients with diagnosed VTE after 72 h from admission were excluded from the study. All variables and markers were documented or calculated upon admission. Death and time to death (in days) were documented at the hospital site where it occurred (COVID-19 ward or COVID-19-ICU). Of note, all patients hospitalized for COVID-19 in our hospital received thromboprophylaxis with low-molecular-weight heparin as per the national and international guidelines of COVID-19 disease management. The study outcomes included the length of stay (LoS) >7 days, intubation, and death during hospitalization.

### 2.2. Statistical Analysis

#### 2.2.1. Propensity Score Matching and Statistical Analysis

Statistical analyses were performed with the use of the Statistical Package for Social Sciences (SPSS) software (SPSS, IBM corp., Armonk, NY, USA), provided by the University of Ioannina. Continuous numeric variables are presented as the mean ± standard deviation (Std deviation). Categorical variables are presented as the total number (N) and percentage (%). Propensity score matching (PSM) was achieved through the FUZZY ver. 2.0.1 (Python script) extension package for SPSS to mitigate bias resulting from confounding variables. Statistical analysis of the patients’ baseline characteristics was performed, and based on that, we proceeded with propensity score matching. The confounding variables used for the matching process were gender, age, morbid obesity, medical history of coronary artery disease (CAD), diabetes mellitus (DM), arterial hypertension (AH), dyslipidemia, cancer, and smoking. The matching ratio in the study was 1 case to 3 controls, with a match tolerance of 0.01. Analyses were performed comparing distinct groups of patients. The exposure group was defined as the group of patients with a diagnosis of VTE (PE and/or VTIB). The control group was defined as the group of patients without a diagnosis of VTE. A chi-squared test was applied to compare the categorical variables between the study groups, the Mann–Whitney and Kolmogorov–Smirnov tests were used for continuous and ratio data, respectively, and binary logistic regression was used, with the outcomes as dependent variables. Our data were found to be not normally distributed, hence the selection of the above statistical tests. Two-tailed significance was defined as a *p*-value < 0.05.

#### 2.2.2. Logistic Regression Using Python and Machine Learning Algorithmic Analysis

We trained multivariate prognostic models using binary logistic regression and machine learning (ML) algorithms. The covariates used in these models were selected based on the results of the univariate binary logistic regression analysis and physicians’ suggestions. For the logistic regression (LR) analysis, we used kNN Imputer to fill in the missing values and hyperparameter tuning to optimize the model. The dataset was divided into 80% training and 20% test sets, and the algorithm was run for 50 iterations. The odds ratios (ORs) and the accompanying *p*-values presented for this model concern one iteration.

XGBoost and AdaBoost are two of the most robust ML models that are successfully utilized to predict medical cases. The XGBoost model, which stands for extreme gradient boosting, is a method of ensemble learning that integrates numerous decision trees [49]. It is very useful for the prediction of medical events due to its ability to handle the complicated interactions and patterns of medical data. Thus, it offers accurate results for disease diagnosis, risk stratification, and medical outcome prediction. Moreover, Freund and Schapire came up with the equally useful AdaBoost algorithm in 1997 [50]. Its excellent compatibility, quick speed, and low complexity make it a popular choice. Clinicians may find AdaBoost to be a useful tool in medical event prediction, as it enhances the model’s sensitivity and specificity. AdaBoost may be used for a variety of tasks, including finding uncommon diseases, spotting disorders early, and choosing the best diagnostic procedures.

The machine learning models used were extreme gradient boost (XGB), adaptive boost (AdaBoost), and LR. During data pre-processing, we utilized kNN Imputer, hyperparameter tuning, and down-sampling to achieve a 1:1 case–control ratio (random selection of controls from the matched population), which is optimal for these applications. In order to assess the performance of these models, we calculated the mean values of the area under the curve (AUC), sensitivity, and specificity of each model across the aggregate of the runs. The performance metrics mentioned are the mean values across these runs. These models were not developed for an outcome of LoS > 7 days due to the extreme difference in group size.

## 3. Results

### 3.1. Study Population Characteristics

Throughout the study period, a total of 1186 consecutive patients were initially included. After propensity score matching in the entire cohort, a total of 532 eligible patients were identified and included in the final analyses. A total of 133 patients diagnosed with a thrombotic event (PE or VTIB) either upon admission or within the initial 72 h after admission were allocated to the VTE group (exposure group). In the non-VTE (control) group, a total of 399 patients were included. The flow diagram of the study is presented in Figure 1.

The patients’ mean age in the entire cohort was 55.4 years. The patients’ age in the VTE group was higher than those in the non-VTE group. Male patients had higher representation than females across all groups.

The prevalence of comorbidities was as follows: arterial hypertension (AH) and dyslipidemia prevailed as the most frequent among patients across all groups. Smoking was more frequent in the group of patients with VTE, while the prevalence of morbid obesity was comparable between the two groups. The baseline (upon admission) demographic characteristics and comorbidities of the entire study population are summarized in Table 1.

### 3.2. Inflammatory Markers in Patients with and without VTE

Fibrinogen and d-dimer levels were significantly higher in patients with VTE compared to patients without VTE (562.2 vs. 516.8 mg/dL, *p* = 0.01, and 1.7 vs. 1.1 μg/mL, *p* = 0.03, respectively). Lactate dehydrogenase (LDH) levels were also higher in patients with VTE compared to patients without VTE (346.0 vs. 318.5 IU/L, *p* = 0.01). Similarly, procalcitonin and C-reactive protein (CRP) levels were higher in the VTE group of patients compared to the non-VTE group (0.4 vs. 0.2 ng/mL, *p* < 0.01 and 81.3 vs. 65.0 mg/L, *p* < 0.01, respectively).

High-density lipoprotein cholesterol (HDL-C) levels were lower in patients with VTE compared to patients without VTE (34.3 vs. 36.4 mg/dL, *p* = 0.13), while triglycerides (TRG) were higher in the patient group with VTE (123.4 vs. 110.5 mg/dL, *p* < 0.01). The ratios of CRP/HDL-C and TRG/HDL-C were significantly higher in the VTE patient group compared to the non-VTE group (2.3 vs. 2.1, *p* = 0.05 and 3.7 vs. 3.3, *p* = 0.01). Similarly, the triglyceride–glucose (TyG) index levels were higher in the VTE group compared to the non-VTE group (8.8 vs. 8.6, *p* = 0.10).

Neutrophil, lymphocyte, and platelet counts were higher in the group of patients with VTE, though no significant difference was found between the groups. Similarly, no significant difference was found in the neutrophil-to-lymphocyte (Neut/Lymph), neutrophil-to-HDL-C (Neut/HDL-C), or lymphocyte-to-HDL-C (Lymph/HDL-C) ratios. Of note, the latter two ratios were higher in the VTE group compared to the non-VTE group of patients (167.9 vs. 160.7, *p* = 0.39, and 54.3 vs. 30.9, *p* = 0.25, respectively). Table 2 summarizes the compared results of the inflammation markers between the groups.

### 3.3. Disease Severity and Outcome in Patients with and without VTE

The patients with VTE exerted a longer duration of symptoms prior to hospitalization compared to the patients without VTE (8.1 vs. 6.6 days, *p* = 0.02). Similarly, the patients in the VTE group had significantly more extensive pulmonary disease (CTBoD) compared to the group of patients without VTE (61.1 vs. 48.0%, *p* < 0.01). On the other hand, the latter required shorter hospitalization (10.1 vs. 14.2 days, *p* < 0.01).

The incidence of all three outcomes was higher in the VTE group of patients vs. the non-VTE group, overall. A greater number of patients was prone to longer hospitalization beyond 7 days in the VTE group vs. the non-VTE group (73.7% vs. 52.4% of patients). Similarly, the rates of intubation and death were higher in patients with VTE compared to patients without VTE (11.3% vs. 4.8% and 12.8% vs. 4.3% of patients, respectively). The indices of disease severity and outcomes in each patient group are summarized in Table 3.

### 3.4. Association of Inflammation Markers and Outcomes in the VTE Group of Patients

#### 3.4.1. Univariate Logistic Regression Analysis. Inflammation and Outcomes in Patients with VTE

The patients’ mean age resulted in a higher risk of intubation and death in those with VTE (OR: 1.03, *p* = 0.06 and OR: 1.05, *p* = 0.01, respectively). Lymphocytopenia was associated with an increased risk for all three outcomes (length of hospital stay (LoS) > 7 days, intubation and death) in these patients (OR: 3.08, 3.81, and 3.35, respectively, all *p* < 0.05). Similarly, the Neut/Lymph ratio was also associated with a higher risk of intubation and death (OR: 1.10 and 1.09, respectively, all *p* = 0.01).

Lower levels of the pO_2_/FiO_2_ ratio (PFR) also increased the risk of intubation (OR: 0.99, *p* = 0.05). Extensive lung injury (CTBoD > 50%) was associated with a higher probability of occurrence for all three outcomes (OR: 5.65, 12.29, and 13.48, all *p* ≤ 0.01). In addition, higher TyG index levels were associated with a greater risk of intubation and death (OR: 2.55, *p* = 0.06). The results of the univariate logistic regression analysis in the VTE group of patients are summarized in Table 4.

#### 3.4.2. Multivariate Logistic Regression Analysis Using Machine Learning (ML)

Decreased body mass index (BMI) values and d-dimer levels >2 μg/mL were associated with a higher risk of intubation after imputation (OR: 0.75, *p* = 0.02 and OR: 19.15, *p* = 0.01, respectively). Increased levels of CRP and CTBoD > 50% were associated with an increased risk of death (OR: 5.95, *p* = 0.01 and OR: 6.31, *p* = 0.05, respectively). The results of multivariate logistic regression analysis using ML are summarized in Table 5.

The performance metrics (accuracy, sensitivity, specificity) of each ML model are presented in Table 6. LR is the fittest model according to these metrics, due to exhibiting high accuracy and equally high sensitivity and specificity. The receiver operating characteristic (ROC) curves for each model are presented in Figure 2. AdaBoost exhibited the highest AUC for intubation prognosis, while LR had the highest AUC for death.

## 4. Discussion

In this study, we assessed the association between inflammation markers and the incidence of VTE in hospitalized COVID-19 patients. In our cohort, VTE was observed in 11.2% of the total population. Concerning the patients with VTE, lymphocytopenia and a high Neut/Lymph ratio were associated with a higher incidence of intubation and risk of death. Similarly, the risk of intubation and death was increased in patients of this group with extensive lung injury (CTBoD > 50%).

It has been shown that COVID-19 disease severity increases VTE risk in hospitalized patients (moderate disease OR: 2.79, and severe disease OR: 5.94) [36,51]. The incidence of VTE overall approximately ranges from 1/10 to 1/3 in COVID-19 patients [37,43]. In the meta-analysis by Kunutsor et al., (n = 9249) in COVID-19 patients, VTE, PE, and DVT incidence rates were 18.4%, 13.5%, and 11.8%, respectively [36]. In another meta-analysis (n = 18,093), the incidence rates for the same outcomes were 17.0%, 7.1%, and 12.1%, respectively [37]. Similarly, in the meta-analysis by Tan et al., (n = 64,503), the incidence rates of VTE (overall), PE, and DVT were 14.7%, 7.8%, and 11.2%, respectively [28]. On the other hand, in Kollias et al.’s meta-analysis (n = 6459), the PE and DVT incidence rates were higher (pooled estimate: 32% and 27%, respectively) [52]. Our study shares similar results with most of these studies regarding VTE incidence. In our cohort, VTE (PE and VTIB) occurred in approximately 1/10 of the entire population.

Of note, it has been shown that VTE incidence is higher upon admission screening. Data analysis from 188 hospitals (n = 374,244) has shown that 78.0% of VTE events (n = 17,346) were diagnosed upon admission [53]. Similarly, in our study, 92% of the total VTE events were observed within the first 72 h of hospitalization.

Several pathophysiological mechanisms have been implicated in SARS-CoV-2-associated thromboembolism. Two major interconnected pathways are responsible for micro- and large-vessel thrombosis [54,55]. The hypercoagulable state and immune-mediated thrombosis, resulting in large-vessel embolism and micro-vessel thrombosis, respectively, both exert significant roles in the development of venous thrombotic events in COVID-19 patients [7,54,56,57]. Also, the thrombotic risk is aggravated by the direct activation of platelets through the spike protein of the virus and inflammatory cytokines (IL-1β, IL-6, and IL-8) [56]. Furthermore, viral-induced endothelial dysfunction enhances this thrombotic state as SARS-CoV-2 directly infects ACE2-expressing cells, resulting in endothelial damage in the lungs and vessels [56,57].

Comparable to our results, there are reports of significantly higher fibrinogen and CRP levels in patients with VTE vs. patients without VTE [58]. In the multicenter cohort study by Lee et al. (n = 3531), it was shown that CRP and LDH levels were significantly higher in the group of patients with VTE vs. the non-VTE group (9.8 vs. 7.6 mg/dL, *p* < 0.001 and 438.1 vs. 380.0 IU/L, *p* = 0.036, respectively) [59].

Recent studies have shown that COVID-19 patients exhibit increased blood neutrophil levels and neutrophil extracellular traps (NETs) [60,61]. Neutrophil activation caused by SARS-CoV-2 enhances cell damage, NET formation, and platelet aggregation and adhesion as well [60,61]. Lymphocytopenia in COVID-19 is considered one of the hallmarks of disease severity, and its role in predicting worse outcomes has been shown [62,63]. In our study, the Neut/Lymph ratio was associated with a greater risk of death in the group of patients with VTE. Similarly, in the study by Toori et al., it was shown that the Neut/Lymph ratio > 3.0 was associated with worse outcomes and increased mortality [64].

The TyG index and TRG/HDL-C ratio are markers of insulin resistance that have exhibited an association with disease severity and worse outcomes in COVID-19 patients [[65],[66]，[67]]. Similar results were found in our study, though no statistical significance was found.

Our findings should be interpreted in the context of certain limitations. By design, this was a retrospective observational study, limiting its generalizability. In addition, a major limitation of this study lies in the small sample size of the VTE patient group. Population matching was performed considering only known confounding parameters, which can intervene in the coagulation process. Also, the lack of bedside ultrasonography in the COVID-19 ward could not exclude that patients in the non-VTE group might have had asymptomatic DVT, thus introducing a confounding effect.

## 5. Conclusions

In summary, our results confirm that inflammatory markers are associated with VTE events in hospitalized COVID-19 patients. Of those, specific markers are also associated with poorer outcomes. However, the results of our study should be interpreted with caution and further investigation. Encompassing various populations and variant periods is needed in order to confirm a link between inflammation and outcomes in COVID-19-associated thrombosis.

The definitions and dictionary of variables in the study’s registry can be downloaded at Appendix A.

## Figures and Tables

**Figure 1 diagnostics-13-03477-f001:**
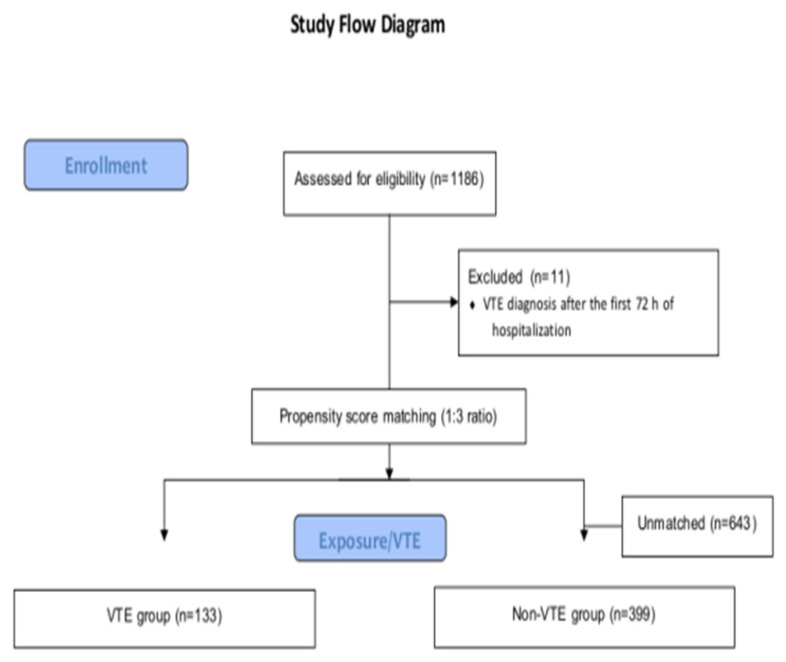
Flow diagram of the study.

**Figure 2 diagnostics-13-03477-f002:**
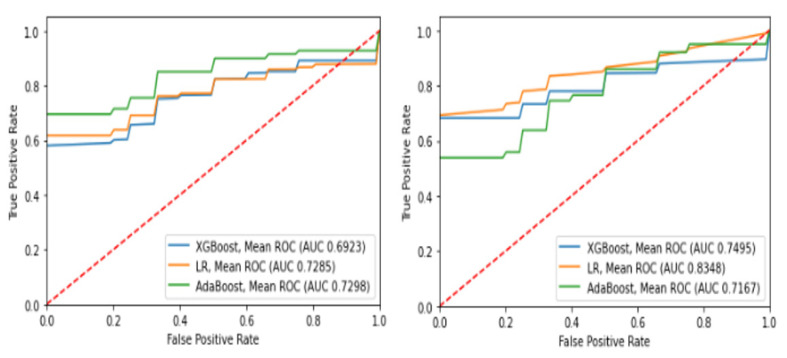
ROC curve of the ML models developed for the outcomes of death (**left**) and intubation (**right**).

**Table 1 diagnostics-13-03477-t001:** Patient characteristics at baseline (admission); data are presented as mean values, cases (n), and percentage (%). BMI: body mass index, VTE: venous thromboembolism, CAD: coronary artery disease, DM: diabetes mellitus, AH: arterial hypertension, CKD: chronic kidney disease.

	Total (N = 532)		VTE Group (N = 133)		Non-VTE Group (N = 399)	
	(n)	(%)	(n)	(%)	(n)	(%)
Demographics						
Gender (male/female)	298/234	56.0/44.0	76/57	57.1/42.9	222/177	55.6/44.4
Age (mean: years)	55.4	-	60.9	-	53.5	-
BMI (mean: kg/m^2^)	29.1	-	30.1	-	29.1	-
Vaccination	104	21.0	14	11.3	90	24.2
Comorbidities—risk factors						
AH	136	25.6	66	49.6	70	17.5
Dyslipidemia	83	15.6	48	36.1	35	8.8
DM	39	7.3	24	18.0	15	3.8
CAD	40	7.5	15	11.3	25	6.3
Thyroid disease	61	12.2	10	7.9	51	13.5
Pulmonary disease	21	3.9	6	4.5	15	3.8
Autoimmune disease	33	6.2	6	4.5	27	6.8
CKD	13	2.4	2	1.5	11	2.8
Cancer	26	4.9	6	4.5	20	5.0
Dementia	12	2.4	2	1.6	10	2.5
Smoking	15	2.8	15	11.3	0	0
Morbid obesity	46	8.6	11	8.3	35	8.8

**Table 2 diagnostics-13-03477-t002:** Inflammation markers in the entire cohort and between-group comparisons; data are presented as mean values, standard (Std) deviation, and SI units. VTE: venous thromboembolism, aPTT: activated partial thromboplastin time, LDH: lactate dehydrogenase, IL-6: interleukin-6, T-C: total cholesterol, TRG: triglycerides, HDL-C: high-density lipoprotein cholesterol, LDL-C: low-density lipoprotein cholesterol, Neut: neutrophils, Lymph: lymphocytes. * *p*-value (Mann–Whitney and Kolmogorov–Smirnov tests); non-VTE compared to VTE group (significance; *p* < 0.05).

Groups		Total (n = 574)		VTE Group (n =144)		Non-VTE (n =430)		Mann–Whitney Test
Variables	Units	Mean	Std. Deviation	Mean	Std. Deviation	Mean	Std. Deviation	*p*-Value *
Neutrophils (count)	#/μL	5355	3321	5664	3691	5252	3186	0.45
Lymphocytes (count)	#/μL	1219	2081	1452	3992	1141	657	0.93
Platelet count	#/μL	209,814	82,627	212,267	91,746	208,986	79,423	0.89
Fibrinogen	mg/dL	529.3	147.1	565.2	119.6	516.8	153.8	0.01
D-dimers	μg/mL	1.3	2.4	1.7	3.4	1.1	2.0	0.03
aPTT	sec	34.1	10.9	35.2	15.8	33.6	8.4	0.44
Ferritin	ng/mL	453.7	536.9	507.4	629.9	434.8	499.8	0.22
LDH	IU/L	325.4	132.2	346.0	128.1	318.5	133.0	0.01
IL-6	pg/mL	51.2	132.4	51.3	90.7	51.1	146.3	0.91
Procalcitonin	ng/mL	0.3	1.2	0.4	1.4	0.2	1.2	<0.01
CRP	mg/L	69.1	71.9	81.3	76.6	65.0	69.8	<0.01
T-C	mg/dL	150.4	41.1	152.4	38.1	149.8	42.0	0.33
TRG	mg/dL	113.7	63.4	123.4	60.6	110.5	64.0	<0.01
HDL-C	mg/dL	35.9	9.8	34.3	8.53	36.4	10.1	0.13
LDL-C	mg/dL	89.8	33.9	96.0	36.1	88.0	33.0	0.09
Novel markers								
Neut/Lymph ratio	-	6.2	6.1	6.2	5.7	6.2	6.3	0.43
Neut/HDL-C ratio	-	162.4	172.7	167.9	113.1	160.7	187.5	0.39
Lymph/HDL-C ratio	-	36.4	91.8	54.3	185.2	30.9	19.1	0.25
TRG/HDL-C ratio	-	3.4	2.3	3.7	2.0	3.3	2.4	0.01
CRP/HDL-C ratio	-	2.1	2.4	2.3	2.0	2.1	2.5	0.05
TyG index	-	8.7	0.6	8.8	0.6	8.6	0.5	0.10

**Table 3 diagnostics-13-03477-t003:** Indices of disease severity and incidence of outcomes; data are presented as mean values, standard (Std) deviation, and SI units. BMI: body mass index, pO_2_: partial pressure of O_2_, FiO_2_: fraction of inspired O_2_, CT: computed tomography, PFR; PO_2_/FiO_2_ ratio, VTE: venous thromboembolism, LoS; length of hospital stay. * *p*-value (Mann–Whitney and Kolmogorov–Smirnov tests); non-VTE compared to VTE group (significance; *p* < 0.05).

Groups		Total (n = 532)		VTE Group (n =133)		Non-VTE Group (n =399)		Mann–Whitney Test
Variables	Units	Mean	Std. Deviation	Mean	Std. Deviation	Mean	Std. Deviation	*p*-Value *
Duration of symptoms	days	7.0	4.6	8.1	5.8	6.6	4.1	0.02
PFR	-	281.4	115.6	263.2	100.9	287.7	119.8	0.02
CTBoD	%	52.6	24.8	61.1	19.6	48.0	26.1	<0.01
Days to death	days	23.8	16.5	26.3	17.8	21.4	15.2	0.49
Days of hospitalization	days	11.1	9.2	14.2	10.5	10.1	8.4	<0.01
Outcomes	-	(n)	(%)	(n)	(%)	(n)	(%)	-
LoS (>7)	days	307	58.1	98	73.7	209	52.4	-
Intubation	-	34	6.4	15	11.3	19	4.8	-
Death	-	34	6.4	17	12.8	17	4.3	-

**Table 4 diagnostics-13-03477-t004:** Univariate logistic regression analysis of inflammation markers and outcomes in the VTE group of patients. OR: odds ratio, VTE: venous thromboembolism, Neut/Lymph ratio: neutrophil-to-lymphocyte ratio, CRP: C-reactive protein, IL-6: interleukin-6, LDH: lactate dehydrogenase, aPTT: activated partial thromboplastin time PFR: pO_2_/FiO_2_ ratio, CTBoD: CT burden of disease, TyG: triglyceride–glucose index, LoS: length of stay. Variables are presented as continuous and categorical (categorical; if noted: “variable” > “of”).

VTE Group	LoS > 7 Days		Intubation		Death	
	OR	*p*-Value	OR	*p*-Value	OR	*p*-Value
Age	1.01	0.38	1.03	0.06	1.05	0.01
BMI	1.03	0.57	0.85	0.16	0.85	0.16
Duration of symptoms	1.06	0.14	1.07	0.11	1.05	0.19
Neutrophil count	1.00	0.87	1.00	0.26	1.00	0.28
Lymphocyte count	1.00	0.85	1.00	0.59	1.00	0.54
Platelet count	1.00	0.96	1.00	0.89	1.00	0.99
Leukocytosis	0.65	0.40	0.79	0.77	0.67	0.61
Lymphocytopenia	3.08	0.01	3.81	0.02	3.35	0.03
Thrombocytopenia	1.38	0.52	0.22	0.15	0.20	0.13
Neut/Lymph ratio	1.08	0.10	1.10	0.01	1.09	0.01
Neut/Lymph ratio > 3.1	0.73	0.52	5.33	0.11	6.24	0.08
Fibrinogen	1.00	0.99	1.00	0.48	1.00	0.41
Fibrinogen > 600	0.90	0.87	1.26	0.75	1.66	0.46
D-dimers	0.96	0.51	1.07	0.28	1.06	0.30
D-dimers > 2	0.32	0.04	2.93	0.10	2.63	0.14
LDH	1.00	0.23	1.00	0.06	1.00	0.14
LDH > 230	1.11	0.84	0.56	0.42	0.20	0.20
aPTT	1.01	0.64	0.99	0.97	0.99	0.78
IL-6	0.99	0.12	1.00	0.56	1.00	0.18
IL-6 > 24	1.48	0.41	0.71	0.64	0.70	0.60
Ferritin	1.00	0.60	1.00	0.17	1.00	0.24
Ferritin > 335	0.94	0.90	0.79	0.72	0.82	0.74
Procalcitonin	0.88	0.45	0.89	0.79	0.97	0.93
Procalcitonin > 0.5	0.82	0.70	1.29	0.70	1.66	0.42
CRP	1.00	0.32	1.00	0.62	1.00	0.55
CRP > 100	1.18	0.70	0.75	0.65	0.87	0.82
PFR	0.99	0.13	0.99	0.05	0.99	0.12
PFR < 150	4.16	0.06	3.04	0.10	2.32	0.19
PFR < 300	1.10	0.80	3.64	0.10	2.68	0.14
CTBoD > 50%	5.65	<0.01	12.29	0.01	13.48	0.01
Troponin	0.99	0.42	1.01	0.06	1.00	0.07
TRG/HDL-C ratio	1.28	0.14	1.24	0.11	1.12	0.39
TRG/HDL-C ratio > 2.5	2.38	0.11	3.11	0.29	0.93	0.93
CRP/HDL-C ratio	1.08	0.52	0.79	0.29	0.80	0.27
TyG index	1.18	0.69	2.55	0.06	1.77	0.22
8.7 < TyG < 9.1	1.85	0.35	0.76	0.82	0.35	0.37
TyG > 9.1	0.70	0.61	6.57	0.09	6.57	0.09
TyG > 9.1 (compared to <8.7)	1.29	0.66	5.00	0.06	2.35	0.22

**Table 5 diagnostics-13-03477-t005:** Multivariate logistic regression models using ML algorithms. OR: odds ratio, VTE: venous thromboembolism, Neut/Lymph ratio: neutrophil-to-lymphocyte ratio, CRP: C-reactive protein, LDH: lactate dehydrogenase, PFR: pO_2_/FiO_2_ ratio, CTBoD: CT burden of disease, TyG: triglyceride–glucose index. Variables are presented as continuous and categorical (categorical; if noted: “variable” > “of”).

VTE Group	Intubation		VTE Group	Death	
	OR	*p*-Value		OR	*p*-Value
Gender	1.15	0.89	Gender	1.33	0.69
Age	1.05	0.15	Age	1.06	0.07
BMI	0.75	0.02	BMI	1.01	0.89
Duration of symptoms	1.08	0.24	Duration of symptoms	1.74	0.26
Lymphocytopenia	4.12	0.19	Lymphocytopenia	4.19	0.08
Neut/Lymph ratio > 3.1	1.10	0.28	Neut/Lymp ratio	1.31	0.82
D-dimers > 2	19.15	0.01	D-dimers > 2	0.52	0.94
Procalcitonin > 0.5	1.01	0.06	CRP	5.95	0.01
PFR < 300	0.99	0.57	PFR	1.70	0.49
CTBoD > 50%	3.86	0.30	CTBoD > 50%	6.31	0.05
TyG > 9.1	0.06	0.06	TRG/HDL-C > 2.5	0.91	0.91

**Table 6 diagnostics-13-03477-t006:** Performance metrics of ML prognostic models developed for the outcomes of intubation and death using features presented in Table 5.

Performance Metrics	Intubation			Death		
	XGBoost	LR	AdaBoost	XGBoost	LR	AdaBoost
Accuracy	0.73	0.82	0.70	0.69	0.72	0.71
Sensitivity	0.73	0.80	0.66	0.71	0.70	0.75
Specificity	0.76	0.86	0.76	0.67	0.75	0.70

## Data Availability

Data are not shared due to privacy issues.

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
