# Peer review of "Inflammation and Venous Thromboembolism in Hospitalized Patients with COVID-19"

_diagnostics, 2023, doi:10.3390/diagnostics13223477_

Round 1
Reviewer 1 Report
Comments and Suggestions for Authors
Liontos et al. conducted a single center retrospective cohort study to investigate the association between inflammation and outcomes in COVID-19 hospitalized patients with VTE.
The paper is of interest, however it has a quite limited sample size some issues should be addressed before the paper can be accepted for publication.
Page 3, page 114: any data regarding thrombophylaxis during hospital stay , which could influence the occurrence of VTE should be reported if available.
Page 3, lines 122-123: suspected DVT could not be confirmed by ultrasonography, as a result only PE could be diagnosed by CT. The title should reflect this issue as : Inflammation and Pulmonary Embolism in Hospitalized Patients with COVID-19. I also would suggest to change VTE to PE throughout the paper.
Statistical analysis: it is unclear why machine learning was applied to a limited sample size ( certainly not big data the total sample being a total of 532 patients). In addition the terms Extreme Gradient Boost (XGB), Adaptive 155 Boost (AdaBoost), should be briefly described for clarity for non expert readers.
Author Response
REVIEWER 1
Comments and Suggestions for Authors
Liontos et al. conducted a single center retrospective cohort study to investigate the association between inflammation and outcomes in COVID-19 hospitalized patients with VTE.
The paper is of interest, however it has a quite limited sample size some issues should be addressed before the paper can be accepted for publication.
Question #1
Page 3, line 114: any data regarding thrombophylaxis during hospital stay , which could influence the occurrence of VTE should be reported if available.
Answer to question #1
We would like to thank the reviewer for the comment. All patients hospitalized for COVID-19 in our hospital received throboprophylaxis with low molecular weight heparin as per national and international guidelines of COVID-19 disease management. We have added this information under section 2.1. Study design and data extraction.
Question #2
Page 3, lines 122-123: suspected DVT could not be confirmed by ultrasonography, as a result only PE could be diagnosed by CT. The title should reflect this issue as : Inflammation and Pulmonary Embolism in Hospitalized Patients with COVID-19. I also would suggest to change VTE to PE throughout the paper.
Answer to question #2
We would like to thank the reviewer for the comment. We agree that VTE is a more general term regarding venous thromboembolism and indeed we could not confirm DVT, however the reviewer should agree that we have dealt with to distinct cases of VTE, namely Pulmonary Embolism and Vascular Tree-in-bud. In order to increase clarity between these two we would like to retain the title as it is and similarly the term VTE in our manuscript.
Question #3
Statistical analysis: it is unclear why machine learning was applied to a limited sample size ( certainly not big data the total sample being a total of 532 patients). In addition the terms Extreme Gradient Boost (XGB), Adaptive 155 Boost (AdaBoost), should be briefly described for clarity for non expert readers.
Answer to question #3
We thank the reviewer for the comment. In the field of healthcare, algorithms based on machine learning have shown to be useful tools with great promise, especially when it comes to predicting clinical events and results, as demonstrated by COVID-19. Machine learning (ML) is a branch of artificial intelligence (AI) that creates computer programs that can carry out activities that typically require human intelligence (https://pubmed.ncbi.nlm.nih.gov/37177382/). The goal of the popular field of machine learning technology is to create a computer system that can mimic human intelligence. It is possible to use machine learning in the healthcare industry as it can provide better answers to medical issues, due to its many capabilities. (https://doi.org/10.3390/encyclopedia1010021 ).
Researchers and medical doctors may extract insightful information and make well-informed decisions by using these models, which play an important role in leveraging the large and intricate datasets produced by medical centers. Nevertheless, the machine learning models have also been applied to databases with fewer samples (https://doi.org/10.1038/s41598-021-86735-9 , https://doi.org/10.1038/s41379-020-00700-x , https://pubmed.ncbi.nlm.nih.gov/33035175/), understanding that their application has a wide use in the data sizes. However, a major disadvantage of smaller samples is reduced power and increased bias compared to larger databases. Furthermore, machine learning models assist in the prediction of serious outcomes referred to COVID-19, such as the progress of disease, deaths and hospitalization (https://doi.org/10.1016/j.imu.2023.101188 , https://doi.org/10.1186%2Fs12911-023-02237-w , https://doi.org/10.1016/j.ibmed.2021.100035). These models analyzed patient data, including demographics, history data, clinical symptoms and comorbidities facilitating in prior warnings and risk assessment. Furthermore, these models are able to detect disease patterns increasing the knowledge of it. They also adjust and evolve possible further available data improving the research. We have added in introduction section a shorter paragraph in page 3 and 4 of our manuscript to increase clarity regarding ML.
Similarly we have added a paragraph in page 4 under the section 2.2.2. Logistic Regression using Python and Machine Learning algorithmic analysis:
XGBoost and AdaBoost are two of the most robust machine learning models that are successfully utilized to predict medical cases. The XGBoost model, which stands for eXtreme Gradient Boosting, is a method of ensemble learning (https://www.ncbi.nlm.nih.gov/pmc/articles/PMC9251895/), that integrates numerous decision trees. It builds simple, brief decision trees by iterations. As a result of its extreme bias, every tree is referred to as a weak learner. XGBoost starts by constructing the first, most basic tree, which performs poorly. Then, it creates a second tree that is taught to anticipate actions that the previous tree—a poor learner—was unable to do. The method generates progressively weaker learners, each of them fixing the preceding tree before the stopping condition (https://doi.org/10.1016/B978-0-323-90548-0.00011-5 ). It is very useful for prediction of medical events, due to the capability of handling the complicated interactions and patterns in medical data. Thus, they offer accurate results for disease diagnosis, risk stratification, and medical outcome prediction. Moreover, Freund and Schapire came up with the extremely useful AdaBoost algorithm in 1997 (https://doi.org/10.1006/jcss.1997.1504 ). Its excellent compatibility, quick speed, and little complexity make it a popular choice. For every characteristic, this method generates a basic weak classifier. It is possible to create a strong classifier with improved performance by integrating the weak classifiers from each iteration. The characteristics that these powerful classifiers utilize have good classification. The algorithm may choose the classifier with the lowest percentage of error while creating the basic classifier. To some extent, this supervision approach can prevent the model from overfitting since it is quite basic, has high classification accuracy, and has acceptable generalization capability (https://doi.org/10.1016/B978-0-12-822830-2.00005-2 ). Clinicians may find AdaBoost to be a useful tool in medical event prediction as it enhances the model's sensitivity and specificity. AdaBoost may be used for a variety of tasks, including finding uncommon diseases, spotting disorders early, and choosing the best diagnostic procedures.
Reviewer 2 Report
Comments and Suggestions for Authors
A manuscript by Liontos et al. entitled “Inflammation and Venous Thromboembolism in Hospitalized 2 Patients with COVID-19” provides evidence that Lymphocytopenia and increased Neutrophil-to-Lymphocyte ratio negatively affect prognosis of COVID-19 disease complicated with venous thromboembolism. The following issues should be addressed:
1. Table 1 does not contain data on statistical analysis of patients’ characteristics. This statistical analysis should be performed to make conclusions concerning the possibility to compare data from these groups and to emphasize the differences in comorbidities.
2. Provide the list of abbreviations used in Table S1 (it should be available in the legend)
3. Page 5 lines 195-201. Please emphasize in the text whether the reduction or elevation of a certain parameter was statistically significant or not.
4. Why Mann-Whitney test was used? Were the data assumed to be not normally distributed?
Author Response
REVIEWER 2
Comments and Suggestions for Authors
A manuscript by Liontos et al. entitled “Inflammation and Venous Thromboembolism in Hospitalized 2 Patients with COVID-19” provides evidence that Lymphocytopenia and increased Neutrophil-to-Lymphocyte ratio negatively affect prognosis of COVID-19 disease complicated with venous thromboembolism. The following issues should be addressed:
Question #1
Table 1 does not contain data on statistical analysis of patients’ characteristics. This statistical analysis should be performed to make conclusions concerning the possibility to compare data from these groups and to emphasize the differences in comorbidities.
Answer to question #1
We thank the reviewer for the comment. We should note that the statistical analysis of patients’ basic characteristics has been performed and based on these results we proceeded to propensity score matching
Question #2
Provide the list of abbreviations used in Table S1 (it should be available in the legend)
Answer to question #2
We thank the reviewer for the comment. As it can be noted in the S1 table 2nd column is the name of the variable and in the third column under the label ‘description” is the full explanation of all mentioned abbreviations, thus they were omitted from the legend as this table serves as a sort of vocabulary.
Question #3
Page 5 lines 195-201. Please emphasize in the text whether the reduction or elevation of a certain parameter was statistically significant or not.
Answer to question #3
We thank the reviewer for the comment. In the mentioned lines by the reviewer all variables discussed are presented as mean values, standard (Std) deviation. All p-values available in the text and table 2 for every metric described shows the significance.
Question #4
Why Mann-Whitney test was used? Were the data assumed to be not normally distributed?
Answer to question #4
We thank the reviewer for the comment. The vast majority of clinical data are not normally distributed. Similarly, our data were found to not be normally distributed as well so Mann-Whitney test was used in the analysis. We have added a comment in page 4 or the manuscript under the section: 2.2.1. Propensity score matching and statistical analysis.
Reviewer 3 Report
Comments and Suggestions for Authors
1) The authors should highlight the novelty of the study and its potential impact on clinical practice (in the introduction section).
2) How did the authors manage the ethical issues and informed consent of patients aged < 18 years (since patients aged > 16 years were enrolled)?
3) Eligibility criteria should be mentioned in the methods section (inclusion and exclusion criteria) - e.g., patients with confirmed VTE; it is not clear if patients with severe SARS-COV2 infections were included or not; excluding criteria are missing (except missing outcome data and VTE after 72 hours). In addition, investigated outcomes should be reported in the methods section.
4) Some important information is missing, such as sample sizes for specific analyses or the timing of data collection. Complete reporting is crucial for readers to understand the study.
5) The authors to clarify the rationale for using machine learning and its relevance to the research (in the introduction or discussion sections).
Author Response
REVIEWER 3
Comments and Suggestions for Authors
Question #1
The authors should highlight the novelty of the study and its potential impact on clinical practice (in the introduction section).
Answer to question #1
We thank the reviewer for the comment. As noted in page 3 on the last paragraph of introduction section, data regarding the association between markers of inflammation and VTE occurrence in COVID-19 are limited. In addition, only a few clinical studies have assessed the role of these markers in COVID-19-related VTE outcomes. The aim of this study was to investigate the association between inflammation and outcomes in COVID-19 hospitalized patients with VTE to enlighten the current available evidence in this group of patients.
Question #2
How did the authors manage the ethical issues and informed consent of patients aged < 18 years (since patients aged > 16 years were enrolled)?
Answer to question #2
We would like to thank the reviewer for the comment. It was a typing mistake and we corrected it accordingly, all patients were ≥18 years old. Patient consent form was waived due to the retrospective study design and the anonymization of the database that was used.
Question #3
Eligibility criteria should be mentioned in the methods section (inclusion and exclusion criteria) - e.g., patients with confirmed VTE; it is not clear if patients with severe SARS-COV2 infections were included or not; excluding criteria are missing (except missing outcome data and VTE after 72 hours). In addition, investigated outcomes should be reported in the methods section.
Answer to question #3
We thank the reviewer for the comment. As noted in section 2.1. in study design and data extraction, we have included all hospitalized patients aged ≥18 years, with a positive RT-PCR test, independently of COVID-19 disease severity. Patients with missing data on outcomes were excluded. In addition, patients diagnosed VTE after 72 hours from admission were excluded from the study. We have also added study outcomes that included Length of Stay (LoS) >7 days, intubation and death during hospitalization.
Question #4
Some important information is missing, such as sample sizes for specific analyses or the timing of data collection. Complete reporting is crucial for readers to understand the study.
Answer to question #4
We thank the reviewer for the comment. As noted in page 3 of our manuscript under the section 2.1. Study design and data extraction, all data regarding patient demographics, anthropometric characteristics, medical history, comorbidities, and concomitant medications were documented on admission (baseline characteristics). Similarly laboratory data were acquired upon admission. Total sample size of the study is reported in page 4 and 5 of the manuscript under the section of study population: Throughout the study period, a total number of 1,186 consecutive patients were initially included. After propensity score matching in the entire cohort, a total of 532 eligible patients were identified and included in the final analyses. A total of 133 patients, diagnosed with a thrombotic event (PE or VTIB) either upon admission or within the initial 72 hours after admission were allocated in the VTE group (exposure group). In the non-VTE (control) group, a total of 399 patients were included. The flow diagram of the study is presented in Figure 1 of the manuscript. Sample sizes are reported for each group of patients in each table (header of table) of the manuscript respectively.
Question #5
The authors to clarify the rationale for using machine learning and its relevance to the research (in the introduction or discussion sections).
Answer to question #5
We thank the reviewer for the comment. As with a previous reviewer’s comment we have added a new paragraph for the rationale of the Machine Learning use in our study (under introduction section):
In the field of healthcare, algorithms based on machine learning have shown to be useful tools with great promise, especially when it comes to predicting clinical events and results, as demonstrated by COVID-19. Machine learning (ML) is a branch of artificial intelligence (AI) that creates computer programs that can carry out activities that typically require human intelligence (https://pubmed.ncbi.nlm.nih.gov/37177382/). The goal of the popular field of machine learning technology is to create a computer system that can mimic human intelligence. It is possible to use machine learning in the healthcare industry as it can provide better answers to medical issues, due to its many capabilities. (https://doi.org/10.3390/encyclopedia1010021 ).
Researchers and medical doctors may extract insightful information and make well-informed decisions by using these models, which play an important role in leveraging the large and intricate datasets produced by medical centers. Nevertheless, the machine learning models have also been applied to databases with fewer samples (https://doi.org/10.1038/s41598-021-86735-9 , https://doi.org/10.1038/s41379-020-00700-x , https://pubmed.ncbi.nlm.nih.gov/33035175/), understanding that their application has a wide use in the data sizes. However, a major disadvantage of smaller samples is reduced power and increased bias compared to larger databases. Furthermore, machine learning models assist in the prediction of serious outcomes referred to COVID-19, such as the progress of disease, deaths and hospitalization (https://doi.org/10.1016/j.imu.2023.101188 , https://doi.org/10.1186%2Fs12911-023-02237-w , https://doi.org/10.1016/j.ibmed.2021.100035). These models analyzed patient data, including demographics, history data, clinical symptoms and comorbidities facilitating in prior warnings and risk assessment. Furthermore, these models are able to detect disease patterns increasing the knowledge of it. They also adjust and evolve possible further available data improving the research. We have added in introduction section a shorter paragraph in page 3 and 4 of our manuscript to increase clarity regarding ML.
Round 2
Reviewer 3 Report
Comments and Suggestions for Authors
The authors took into account the recommendations of the reviewers, modified the manuscript in accordance with their suggestions and thus made its publication possible.